# Interoperable slide microscopy viewer and annotation tool for imaging data science and computational pathology

Chris Gorman[1], Davide Punzo [2], Igor Octaviano [2], Steven Pieper[3], William J. R. Longabaugh[4], David A. Clunie [5], Ron Kikinis[6], Andrey Y. Fedorov[6] ✉ & Markus D. Herrmann [1] ✉

The exchange of large and complex slide microscopy imaging data in biomedical research and pathology practice is impeded by a lack of data standardization and interoperability, which is detrimental to the reproducibility of scientific findings and clinical integration of technological innovations. We introduce Slim, an open-source, web-based slide microscopy viewer that implements the internationally accepted Digital Imaging and Communications in Medicine (DICOM) standard to achieve interoperability with a multitude of existing medical imaging systems. We showcase the capabilities of Slim as the slide microscopy viewer of the NCI Imaging Data Commons and demonstrate how the viewer enables interactive visualization of traditional brightfield microscopy and highly-multiplexed immunofluorescence microscopy images from The Cancer Genome Atlas and Human Tissue Atlas Network, respectively, using standard DICOMweb services. We further show how Slim enables the collection of standardized image annotations for the development or validation of machine learning models and the visual interpretation of model inference results in the form of segmentation masks, spatial heat maps, or image-derived measurements.

Slide microscopy imaging data acquired in biomedical research and clinical pathology practice are increasing in size, dimensionality, and complexity[1] and technological advances in machine learning (ML) and artificial intelligence (AI) promise to exploit large and rich imaging data to drive discoveries and to support humans in data interpretation and in data-informed decision making[2–4]. Unfortunately, sharing of imaging data and reporting of imaging findings are lagging with detrimental effects on the reproducibility of microscopy imaging science[5] and on the clinical adoption of computational pathology[6]. Given their immense size, slide microscopy imaging data sets generally exceed the capacity of local storage and are hence stored on remote servers and accessed over a network. A critical requirement for making data findable, accessible, interoperable, and reusable (FAIR) and for enabling AI

is the availability of standardized metadata and standard application programming interfaces (APIs)[2,7–12]. Web-based slide microscopy viewers, which access imaging data via web APIs, have become key for human-machine collaboration in biomedical imaging data science and computational pathology because they enable human experts to browse through and visually analyze large image datasets for hypothesis generation and testing, to select and annotate images for the development and validation of ML models, and to review and visually interpret image analysis results upon ML model inference[13–15]. However, existing web-based slide microscopy viewers are either monolithic or heavily rely on custom interfaces, which tightly couple client and server components and thereby impede interoperability between image analysis, management, and viewing systems.

[1]Department of Pathology, Massachusetts General Hospital and Harvard Medical School, Boston, MA, USA. [2]Radical Imaging, Boston, MA, USA. [3]Isomics Inc, Cambridge, MA, USA. [4]Institute for Systems Biology, Seattle, WA, USA. [5]PixelMed Publishing LLC, Bangor, PA, USA. [6]Department of Radiology, Brigham and Women's Hospital and Harvard Medical School, Boston, MA, USA. ✉e-mail: afedorov@bwh.harvard.edu; mdherrmann@mgh.harvard.edu

The dependence on custom data formats and metadata elements further introduces interdependencies between analysis and visualization or annotation, requiring both ML models and viewers to customize data retrieval and decoding or data encoding and storage for each system they consume data from or produce data for, respectively[6,8,16–18]. It would thus be desirable for analysis tools and viewers to rely on standard web APIs and data formats to become agnostic to internal implementation details of image management systems, to enable the visualization of image analysis results generated by a broad range of ML models or other computational tools, and to allow for the generation of machine-readable image annotations that can be readily consumed and interpreted by ML models and reused across projects without the need for extensive data engineering.

In this paper we argue that the existing Digital Imaging and Communications in Medicine (DICOM) standard is well suited to address the challenges articulated above and we present a software implementation that demonstrates the practical utility of using the DICOM standard for quantitative tissue imaging in biomedical research as well as in digital and computational pathology. Specifically, we present the web-based viewer and annotation tool Slim, which realizes FAIR principles and enables standard-based imaging data science by leveraging the DICOM standard for management and communication of digital images and related information. DICOM is the internationally accepted standard for medical imaging[19], catering to the healthcare needs of billions of people on a daily basis. DICOM is often considered a radiology standard, but it has been widely adopted across medical domains including dermatology, ophthalmology, and endoscopy for a variety of imaging modalities and finds broad application in preclinical and clinical research[20–23]. In recent years, the standard has been further developed to support slide microscopy[24,25], quantitative imaging[17,22], and machine learning[26,27], and is being adopted internationally for storage, management, and exchange of slide microscopy in diagnostic pathology[28] as well as in research and development[29,30]. While DICOM has been primarily designed for medical imaging in clinical practice and includes established mechanisms for ensuring high image quality required for diagnostic purposes, the standard also has several advantages for biomedical imaging research[17,25,26]. Notably, DICOM specifies profiles for data de-identification that facilitate the use of clinically acquired imaging data sets for research purposes[31,32] and defines digital objects for encoding and communicating image annotations and analysis results[17,22,26,27]. Furthermore, there is an abundance of open-source software libraries and tools that support DICOM[33].

The National Cancer Institute's Imaging Data Commons (IDC) is a DICOM-based research platform for visual exploration and computational analysis of biomedical images in the cloud that aims to integrate heterogeneous imaging data from multiple modalities and diverse projects to facilitate integrative, multi-modal imaging data science[29]. As part of the Cancer Research Data Commons (CRDC), the IDC provides researchers a scalable, cloud-hosted imaging data repository and tools for searching, sharing, analyzing, and visualizing cancer imaging data. Slim serves as the slide microscopy viewer of the IDC and has been tasked to support a wide range of slide microscopy imaging use cases from traditional brightfield microscopy of hematoxylin and eosin (H&E) stained tissue to highly multiplexed tissue imaging using sequential immunofluorescence microscopy on a common, modality-agnostic DICOM-centric cloud infrastructure that is shared with the radiology viewer[34].

In this paper, we showcase Slim's visualization and annotation capabilities for quantitative tissue imaging research and machine learning model development in the context of the IDC using several public IDC collections of The Cancer Genome Atlas (TCGA)[35], the Clinical Proteomic Tumor Analysis Consortium (CPTAC)[36], and the Human Tumor Atlas Network (HTAN)[1] projects. In addition, we demonstrate the ability of Slim to interoperate with commercial digital pathology systems that support DICOM and present a pathway for streamlining the translation of novel image-based methods and tools from research into clinical practice.

## Results

Slim is a web application that runs fully client side in modern web browsers and communicates with servers via standard DICOMweb services[37] to query, retrieve, or store data (Fig. 1). By relying on the open DICOMweb API specification, Slim is independent of a particular server implementation and can interoperate with any server that exposes a DICOMweb API, including a variety of open-source applications[38–40], commercial Picture Archiving and Communication Systems (PACSs) or Vendor Neutral Archives (VNAs) installed at healthcare enterprises around the world[37,41], and cloud services offered by several major public cloud providers (Fig. 1). Slim can simply be configured with the unique resource locator (URL) of a DICOMweb server and its static assets can be served via a generic web server, which makes it straightforward to integrate Slim into digital imaging workflows on premises or in the cloud. The web application can optionally be configured to use the OpenID Connect (OIDC) protocol[42] to authenticate users and obtain authorization to access data through secured DICOMweb API endpoints. In the case of the IDC, Slim is hosted on the Google Cloud Platform and uses the DICOMweb services of the Google Cloud Healthcare API for data communication[29]. To facilitate public access, the official IDC instance of Slim is configured with the URL of a reverse proxy, which is authorized to access DICOM stores via the Cloud Healthcare API and forwards DICOMweb request and response messages between DICOMweb clients and server.

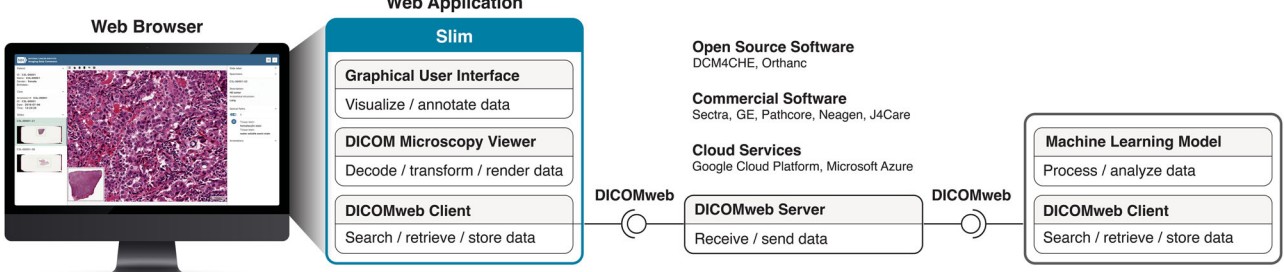

**Fig. 1 | Components of the Slim web application and communication of data using DICOMweb services.** Component diagram showing relevant client and server components and their interfaces. Slim is a single-page application that runs fully client side in a web browser without any custom server side components. All data communication occurs via standard DICOMweb services. The application exposes a graphical user interface for interactive visualization and annotation of image pixel data and internally uses the DICOM Microscopy Viewer and DICOMweb Client libraries for decoding, transforming, and rendering data and for querying, retrieving, and storing data via a DICOMweb interface, respectively.

**Brightfield Slide Microscopy**

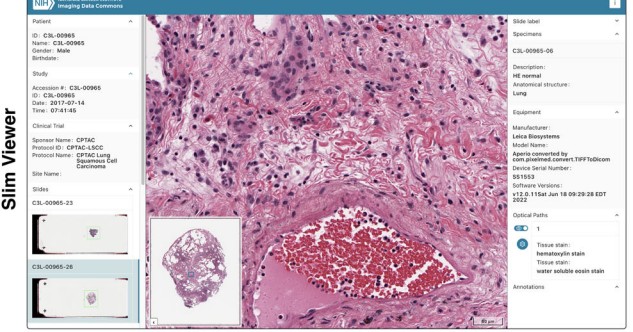

**Fluorescence Slide Microscopy**

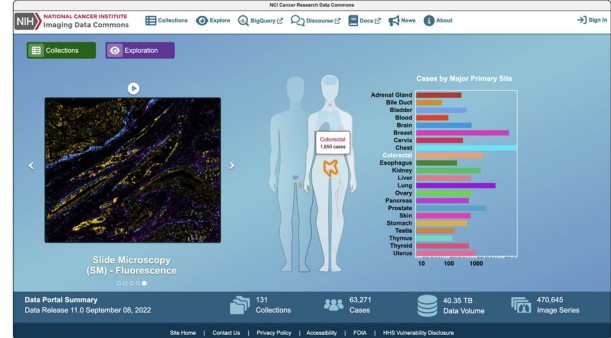

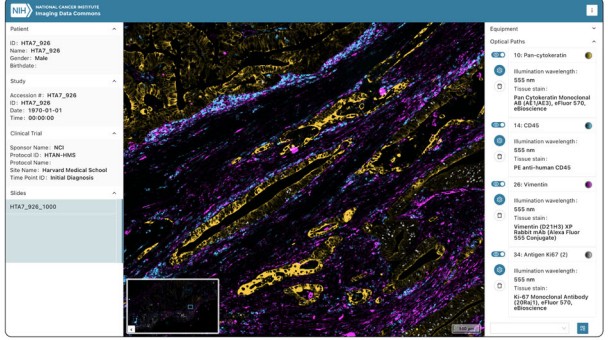

Color Image

Pseudocolor Image

**Fig. 2 | Use of Slim as slide microcopy viewer of the NCI Imaging Data Commons.** Screenshots of the NCI Imaging Data Commons data portal (top) and of the Slim viewer (bottom). Shown are a true color image of a hematoxylin and eosin stained lung squamous cell carcinoma specimen from the Clinical Proteomic Tumor Analysis Consortium (CPTAC) that was acquired via brightfield slide microscopy (left) and a pseudocolor image derived from multiple grayscale images of an immunostained colon carcinoma specimen from the Human Tumor Atlas Network (HTAN) that was acquired via fluorescence slide microscopy (right).

## Enabling interactive visualization of heterogeneous slide microscopy images using standard DICOMweb services

The IDC data portal allows researchers to explore public imaging collections and provides them with links to an instance of Slim through which they can access selected slide microscopy images via DICOMweb and interactively visualize the data in the browser (Fig. 2). The DICOM information model is object-oriented and imaging data are represented and communicated in the form of digital objects, which include various data elements for identifying and describing pertinent information entities. The standard defines different types of information objects for different kinds of imaging data (images, annotations, structured reports, etc.) and the VL Whole Slide Microscopy Image information object definition is used for communication of slide microscopy images. Each image object is composed of a pixel data element, which contains the encoded (usually compressed) pixels of acquired image frames, and additional metadata elements, which describe the organization and structure of the pixel data (number of pixel rows and columns, number of samples per pixel, photometric interpretation, etc.), the image acquisition context and equipment (acquisition date and time, manufacturer, etc.), the imaging study (accession number, etc.), the imaging target (specimen identifier, etc.), or other relevant identifying and descriptive contextual information[25]. The DICOMweb API specification provides separate web resources for metadata, pixel data, and other bulk data elements of DICOM objects, which allow a viewer − or any other application − to search for digital objects and to efficiently access relevant subsets of the data, such as individual image frames, without having to download potentially large objects in their entirety[25]. Slim, as other slide microscopy viewers, requires access to image pixel data in pyramid representation to facilitate seamless navigation across multiple resolutions. However, in DICOM each resolution level is stored as a separate image object and the standard does not require DICOMweb servers to be aware of multi-resolution pyramids or have any knowledge of how individual image instances relate to each other across scale space[25]. In contrast to other viewers, which depend on a specific server implementation with a dedicated image tiling or rendering engine and can thus simply assume a pyramid structure with a constant downsampling factor between resolution levels or a regular tiling across all levels, Slim dynamically determines the multi-resolution pyramid structure from the metadata of DICOM image objects it discovered via DICOMweb. To support a wide range of slide microscopy use cases and enable the display of images acquired by different scanners, Slim has been designed to handle highly variable and sparse pyramid structures with different downsampling factors between levels, different tile sizes per level, missing levels, missing tiles at any level, and a different compression of pixel data at each level. Upon navigation to an imaging study, Slim first searches for available image objects and selectively retrieves the metadata of found objects via DICOMweb to discover how many images exist, how the existing images relate to one another across resolution levels to form a multi-resolution pyramid, and how individual frames are organized within the pixel data of each image with respect to the tile grid at the given resolution level. When the user subsequently pans or zooms, Slim dynamically retrieves the frame pixel data that are needed to fill the viewport tiles from the server via DICOMweb, and then decodes, transforms, and renders the received pixel values on the client for display to the user. To transform stored values into device-independent and accurate display values, Slim implements established DICOM pixel transformations, which are widely used by medical devices for diagnostic purposes in radiology and other clinical domains. It is important to note that these transformations are parametrized by DICOM attributes, which can be included into DICOM image objects by image acquisition devices and then communicated along with other image metadata via DICOMweb.

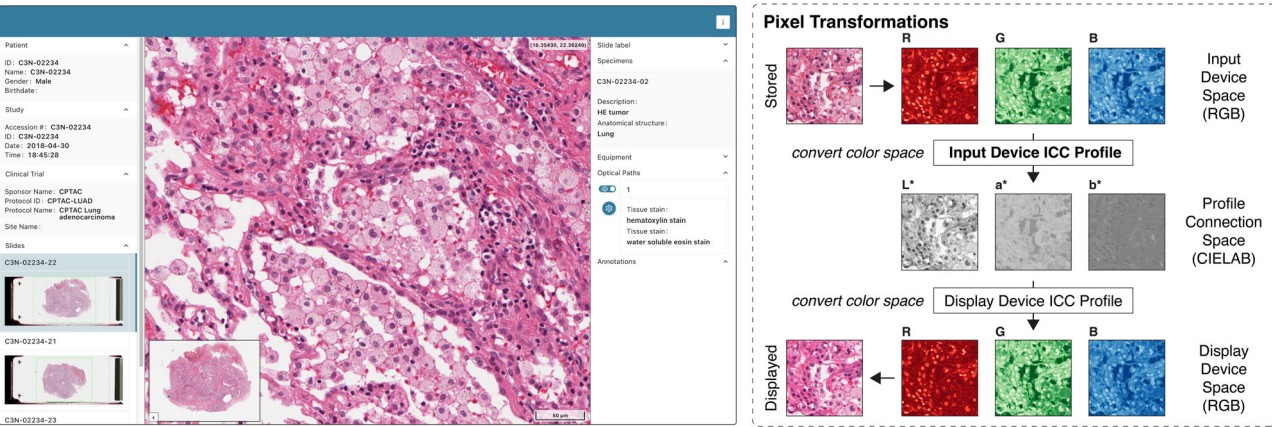

**Fig. 3 | Display of color images acquired via brightfield slide microscopy.**
Screenshot of the Slim user interface displaying a color image of a hematoxylin and eosin stained specimen from the Clinical Proteomic Tumor Analysis Consortium (CPTAC) project that was acquired via brightfield whole slide imaging (left). The viewer applies a sequence of pixel transformations to each retrieved image frame to convert color images from the input device color space into the display device color space (right). Note the subtle but perceptible difference in color of the stored and displayed image frames.

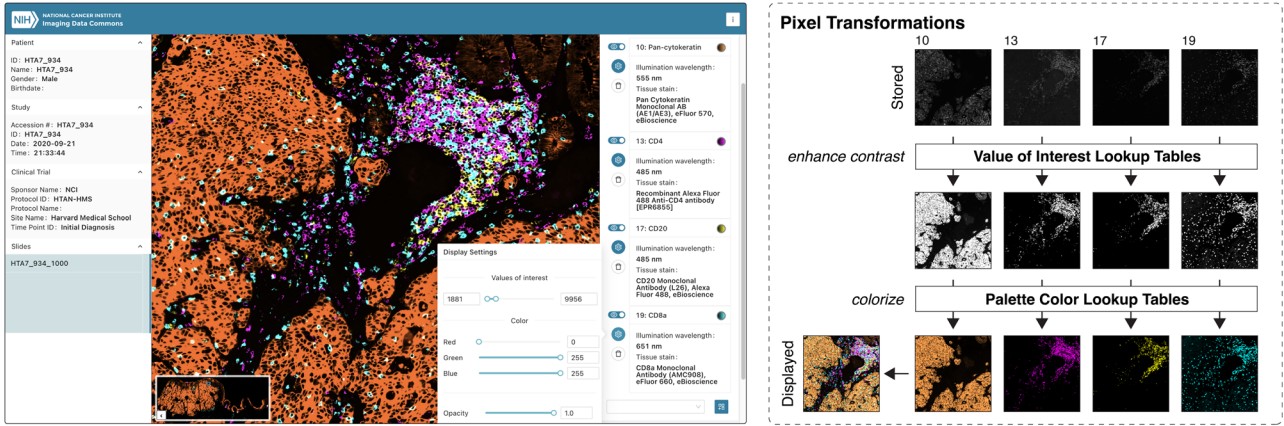

**Fig. 4 | Display of grayscale images acquired via fluorescence slide microscopy.**
Screenshot of the Slim user interface displaying multiple grayscale images of an immunostained colon adenocarcinoma specimen from the Human Tumor Atlas Network (HTAN) that were acquired via cyclic immunofluorescence imaging, where the user manually selected individual image channels for display and adjusted the display settings for each channel (left). The viewer constructed value of interest and palette color lookup tables from the user-provided display settings to contrast enhance and colorize selected grayscale images, respectively, and additively blended the resulting pseudocolor images (right).

## Achieving color accuracy and consistency for visualization of true color images acquired via brightfield microscopy

Color is of critical importance for microscopy imaging, yet accurate and reproducible display of colors is complicated by the fact that different image acquisition and display devices may represent colors in device-specific RGB color spaces with different gamuts, i.e., ranges of chromaticities of the red, green, and blue primaries. The sRGB color space has evolved as the standard color space for the web and is assumed by most web browsers. However, medical imaging devices generally use a device-specific RGB color space that accommodates a wider gamut and allows for more exact digital representation of colors. Display of color images acquired by medical imaging devices via a web browser thus requires controlled conversion of colors between different color spaces[43,44]. Accordingly, regulatory agencies consider color management a critical requirement for virtual microscopy in digital pathology[43,44]. Unfortunately, many existing slide microscopy viewers do not manage colors and fail to reproduce colors[45]. To facilitate accurate display of brightfield microscopy images, Slim implements the DICOM Profile Connection Space Transformation, which uses International Color Consortium (ICC) profiles for color management. An ICC profile specifies a transformation for mapping pixel data from a source color space into a target color space and a rendering intent to handle potential gamut mismatches between source and target color spaces by mapping saturated, out-of-gamut colors in the target color space into the gamut. The ICC color management workflow makes use of two ICC profiles to retain color fidelity: (i) an input device profile for mapping stored pixel values from the RGB color space used by the scanner for image acquisition into a device-independent CIEXYZ or CIELAB color space called the profile connection space (PCS) and (ii) a display device profile for mapping PCS pixel values into the RGB color space used by the viewer for image display (Fig. 3)[46]. DICOM requires scanners to embed an input device ICC profile into DICOM image objects. Slim retrieves the embedded ICC profiles of images via DICOMweb and applies the appropriate input device profile to the decoded pixel values of each image frame and subsequently applies the display device profile. By default, Slim uses a standard display device profile that maps color values into sRGB color space, which is optimized for display of color images via different web browsers on a wide range of monitors. By transforming device-dependent storage values into device-independent display values, Slim facilitates presentation of color images to users as intended by the scanner manufacturer. The extent to which a given input device ICC

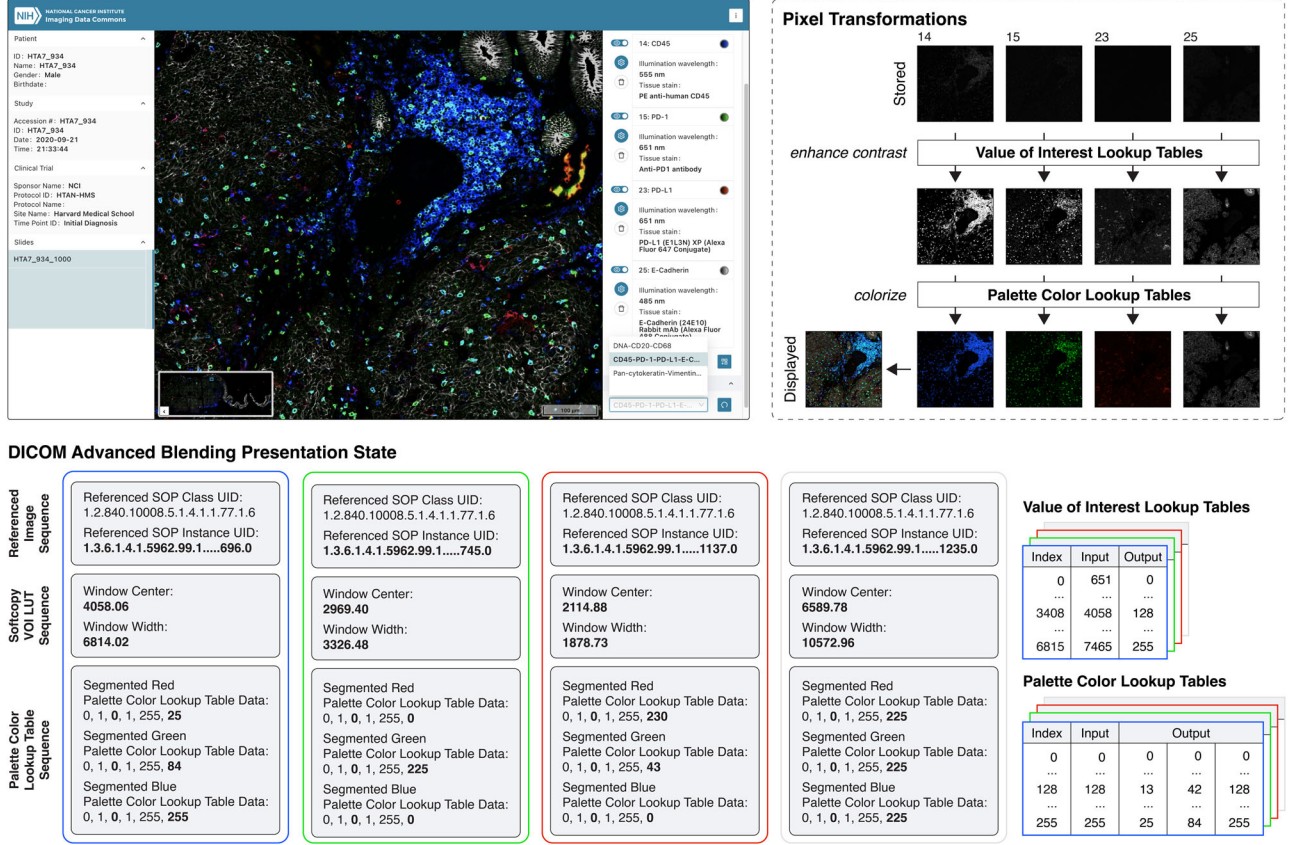

**Fig. 5 | Automatic setting of display parameters using presentation states.** Screenshot of the Slim user interface displaying the same grayscale images as shown in Fig. 4, but where the user chose a presentation state, which automatically selected a set of image channels and adjusted the display settings for each channel (upper left). The viewer constructed value of interest and palette color lookup tables from the provided display settings to contrast enhance and colorize referenced grayscale images, respectively, and additively blended the resulting pseudocolor images (upper right). The reference of selected images, the description of the value of interest (VOI) lookup table (LUT), and the description of the palette color lookup table are encoded in a DICOM Advanced Blending Presentation State instance (bottom). In this case, the VOI LUT represents a linear function that maps 16-bit grayscale values into a window of 8-bit grayscale values and the palette color LUT represents three linear functions that each map grayscale values into 8-bit RGB color values. The VOI window is described by the window center and window width and the palette color ranges for the red, green, and blue channels are described separately via two segments that define the first and last color value. The LUT data are encoded in the DICOM object in binary form, but are shown here as text for the purpose of illustration.

profile will change the visual appearance of an image depends on the color calibration information that the scanner includes in the profile. Some scanners use color spaces with a highly specific gamut and the effect of the input device ICC profile from such a scanner on the image quality can be dramatic (Supplementary Fig. 1)[47].

## Pseudocoloring of grayscale images acquired via fluorescence microscopy and additive blending of pseudocolor images

Display of single or multi-channel grayscale images on color monitors poses a different set of challenges. First, grayscale images are typically acquired with 12-bit or 16-bit cameras and stored as 16-bit unsigned integer values, which usually exceed the bit depth of consumer-grade monitors and the sensitivity of the human visual system to differences in signal intensities, requiring intensity values to be rescaled for display and visual interpretation by human readers[48]. Second, grayscale images are often pseudocolored, which can aid in visual interpretation, but can also sometimes mislead human readers[49]. To facilitate display of fluorescence microscopy images, Slim implements the DICOM VOI LUT Transformation and the DICOM Advanced Blending Transformation, which parametrize the conversion of grayscale images with various bit depths into 8-bit pseudocolor images for display on ordinary color monitors and the additive blending of two or more pseudocolor images into a false color image, respectively. To this end, the viewer first applies a value of interest (VOI) lookup table (LUT) to clip and

rescale stored pixel values of each grayscale image into a normalized range — a process that is referred to as windowing — and then subsequently applies a palette color LUT to map the rescaled grayscale values to RGB values in the range from 0 to 255 (Fig. 4). The viewer can construct the VOI and palette color LUTs directly from user input, allowing users to dynamically adjust contrast and color per channel (Fig. 4). However, if the manufacturer embedded a palette color LUT into DICOM image objects, the viewer will automatically apply the supplied LUT and prevent the user from manually overriding color values to ensure that the images are displayed to human readers as intended by the manufacturer (Supplementary Fig. 2a). For example, a manufacturer may provide a palette color LUT to pseudocolor grayscale images of specimens stained with a fluorescence dye and acquired via fluorescence microscopy such that they appear to pathologists as if they were true color images of specimen stained with a traditional chemical dye and acquired via brightfield microscopy (Supplementary Fig. 2b). If the palette color LUT is created by the viewer based on user input, the color values are already in the display device's color space and no additional color management is necessary. However, if the palette color LUT was generated by another device, the color values are defined in the color space of the input device and they will hence need to be transformed into PCS values using the appropriate input device ICC profile, analogous to true color images. In either case, the user will always retain the ability to change the value of interest range

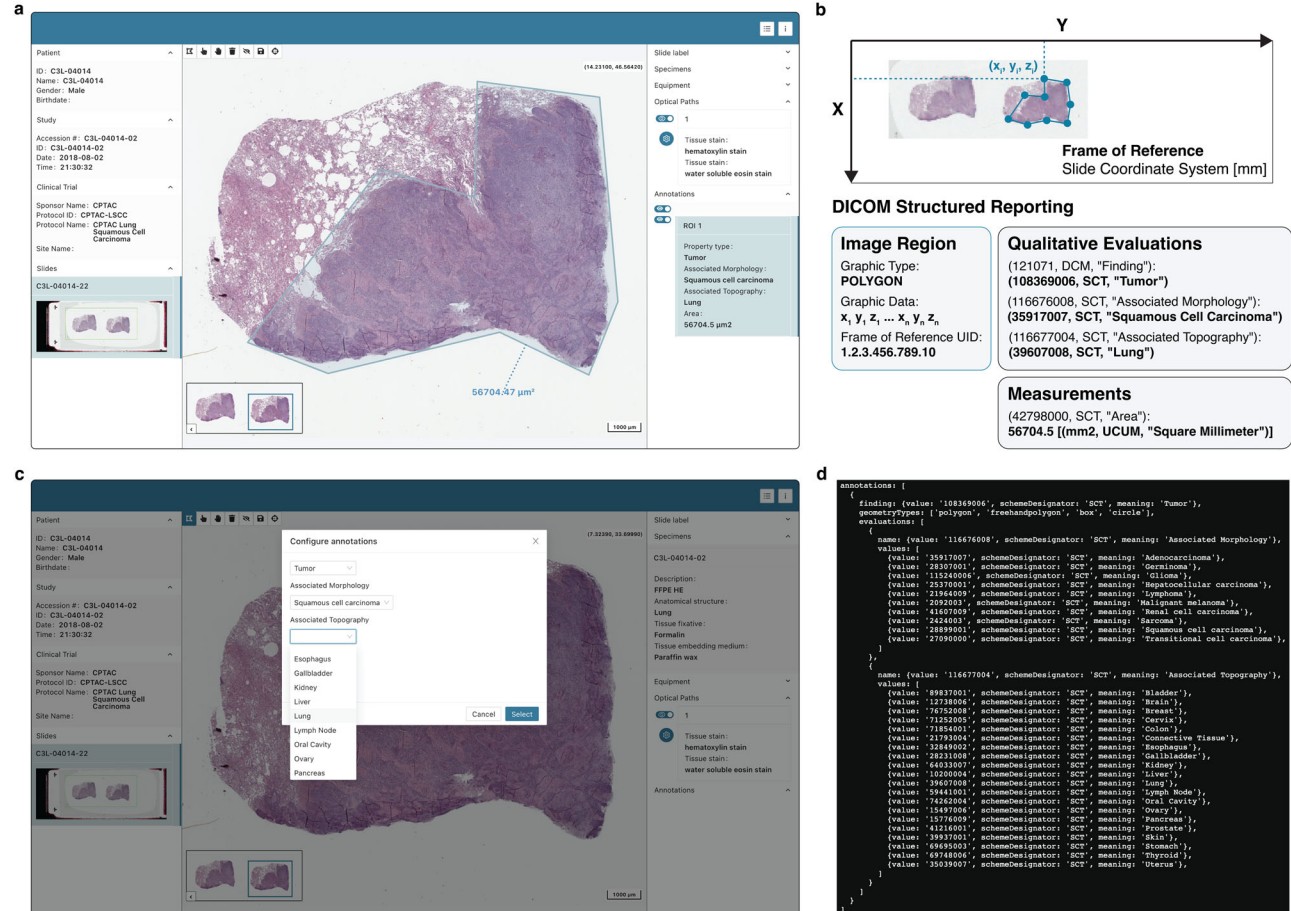

**Fig. 6 | Annotation of image regions of interest. a** Annotation of image regions of interest (ROIs) using DICOM Structured Reporting. **b** Screenshot of the Slim user interface displaying a ROI annotation drawn by a user on images of the Clinical Proteomic Tumor Analysis Consortium (CPTAC) project. Note the user-provided qualitative evaluations and machine-generated size measurements of the ROI that are displayed in the side panel. **c** Screenshot of the Slim user interface displaying a popup window that appears when the user starts to draw regions of interests (ROIs) and that prompts the user to answer questions about the annotated ROIs. The annotation task here is the classification of image regions into different tumor categories, and the user is asked to specify the associated morphology and topography. Note the drop-down menu provides options from which the user can choose an answer without having to enter free text. **d** Section of the JavaScript configuration file that shows the underlying codes that determine both the questions that are posed to users and the list of permitted answers from which users can choose the appropriate one.

and thereby adjust the imaging window to visually explore pixel data along the full bit depth, which is critical for interpretation of tissue stainings that result in signal intensities with a high dynamic range.

### Enabling interpretation of images in context through display of relevant image metadata

A major difference between DICOM and other formats (e.g., TIFF) is that DICOM image objects not only include the pixel data and pixel-related structural metadata (width, height, bit-depth, etc.), but also identifying and descriptive metadata about the patient, the imaging study, the image acquisition equipment, and the imaged tissue specimens, including information about specimen preparation procedures (collection, sampling, fixation, embedding, and staining), which allow human readers to interpret the pixel data in context[50]. These image metadata are indispensable in the clinical setting to allow for unique identification and matching of patients and specimens and to ensure that imaging findings and the final histopathological diagnosis are assigned to the right real-world entity[51]. However, metadata describing the acquisition context are also invaluable in the research setting, especially for immunofluorescence microscopy imaging, where information about the optical path (e.g., the illumination wavelength) and specimen preparation (e.g., the antibody used for immunostaining) are generally needed for unambiguous interpretation of the pixel data[11,25]. Slim obtains metadata of DICOM image objects via DICOMweb without requiring additional, out-of-band connections and displays pertinent identifying and descriptive metadata via the graphical user interface alongside the pixel data (Figs. 1–4). Of note, well-established DICOM profiles for data de-identification provide guidance on how to remove protected health information (PHI) from image objects (e.g., the patient's name and birth date and laboratory accession numbers) for research purposes, while retaining non-PHI that is necessary for data interpretation (e.g., substances used for specimen fixation, embedding, and staining). The DICOM information model further provides attributes specifically for research use, for example clinical trial protocol, sponsor, and site identifiers, which can be invaluable for cohort creation and data set curation for the development of ML models.

### Visual interpretation of highly multiplexed tissue imaging data using DICOM presentation states

As described above, Slim enables users to dynamically select, contrast enhance, and pseudocolor grayscale images of different channels and thereby gives users full control over the display of the pixel data (Fig. 4). This flexibility is often needed to visually explore the data in all

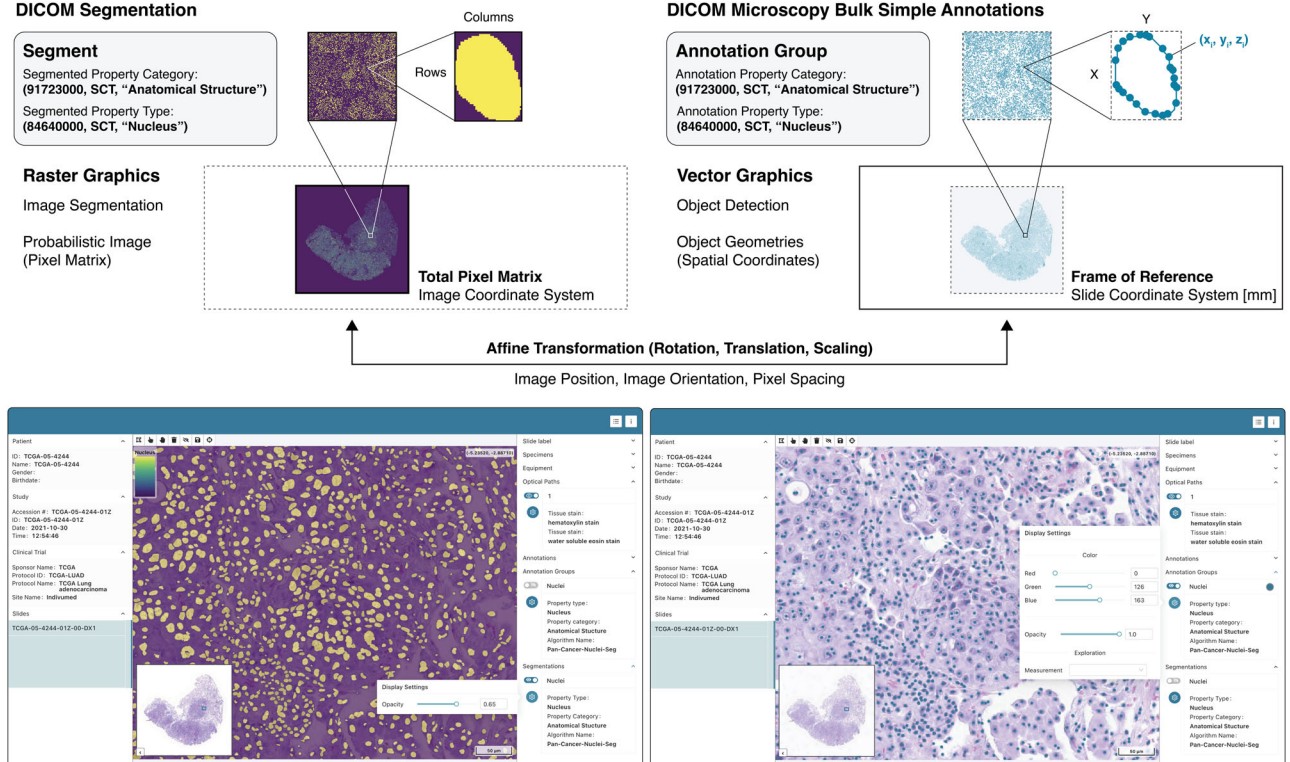

**Fig. 7 | Display of image segmentation and object detection results.** Schematic representation of image segmentation or object detection results as raster or vector graphics, respectively, together with semantic metadata (top) and screenshots of the Slim user interface displaying the data (bottom). Shown are the binary segmentation mask (left) as well as the centroids of detected objects (right), which were derived from slide microscopy images from The Cancer Genome Atlas (TCGA) project. The raster graphic data are represented as a two-dimensional pixel matrix and vector graphic data are represented as spatial coordinates in the three-dimensional slide coordinate system in millimeter units. The slide coordinate system serves as a frame of reference and graphic data may need to be transformed and spatially aligned for overlay onto the source images. The affine transformation is parametrized using appropriate DICOM metadata.

its facets. However, setting all display parameters manually is time consuming and difficult to reproduce exactly, and also demands significant technical and domain knowledge from the user[14]. Visualizing channels of a highly multiplexed tissue imaging data set may easily overwhelm a human reader[14] and selecting appropriate colors is a challenging subject in itself[49,52]. To lower the barrier for scientists and to facilitate the interpretation of complex multi-dimensional imaging data, Slim allows for automatic and persistent configuration of display settings using DICOM presentation states. A DICOM Presentation State (PR) object makes it possible to reference a set of images and define how the referenced images should be presented to human readers in a device-independent manner. DICOM PR objects can be used to persist display settings as data alongside the DICOM image objects and communicate display settings between input and display devices. Slim uses DICOM Advanced Blending Presentation State instances to predefine different combinations of immunofluorescence imaging channels and to rescale and colorize each channel such that an optimal contrast is achieved for visualization of specific anatomical or morphologically abnormal structures, such as tumor cells and different subtypes of tumor infiltrating lymphocytes (Fig. 5). Slim searches for DICOM Advanced Blending Presentation State objects via DICOMweb and allows users to select from the identified presentation states via a dropdown menu, allowing them to dynamically switch between different states (Fig. 5). Importantly, a presentation state does not alter or duplicate stored pixel values, but only specifies how the pixel data should be transformed for presentation. Using presentation states, users retain their ability to dynamically adjust or override display settings as needed, for example to customize the imaging window and enhance image contrast within a specific range of pixel values to highlight biological structures of interest.

## Interactive graphic annotation of slide microscopy images using DICOM structured reporting

In addition to image display, Slim supports image annotation by allowing researchers to draw and label image regions of interest (ROIs), for example to generate a ground-truth for training or validation of ML models. The viewer provides various annotation tools for drawing, labeling, and measuring ROIs using DICOM Structured Reporting (SR)[23,53] and for storing the annotations as DICOM SR documents via DICOMweb. Through the annotation tools, Slim enables users to draw ROIs in the form of vector graphics and to associate measurements and qualitative evaluations with individual ROIs (Fig. 6a). Slim supports all graphic types specified by DICOM SR to represent ROIs in the form of individual points or object geometries such as lines, polygons, circles, and ellipses, which are defined by multiple points. The spatial coordinates of each point are internally tracked in the three-dimensional slide coordinate system in millimeter unit, making them independent of the pixel matrix of an individual image instance and translatable to any image in the same frame of reference across different channels and resolution levels (Fig. 6b). DICOM SR further allows for encoding of qualitative evaluations or measurements using standard coding schemes such as the Systematized Nomenclature of Medicine Clinical Terms (SNOMED CT)[54] to endow annotations with well-defined semantics and to make them readable and interpretable by both humans and machines (Fig. 6b). Each qualitative evaluation and measurement is thereby represented as a name-value pair (or a question-answer pair). While the name is always coded, the representation of the value depends on the data type. In the case of a qualitative evaluation, the value is coded as well, while in the case of a measurement, the value is numeric (an integer or floating-point number) (Fig. 6b).

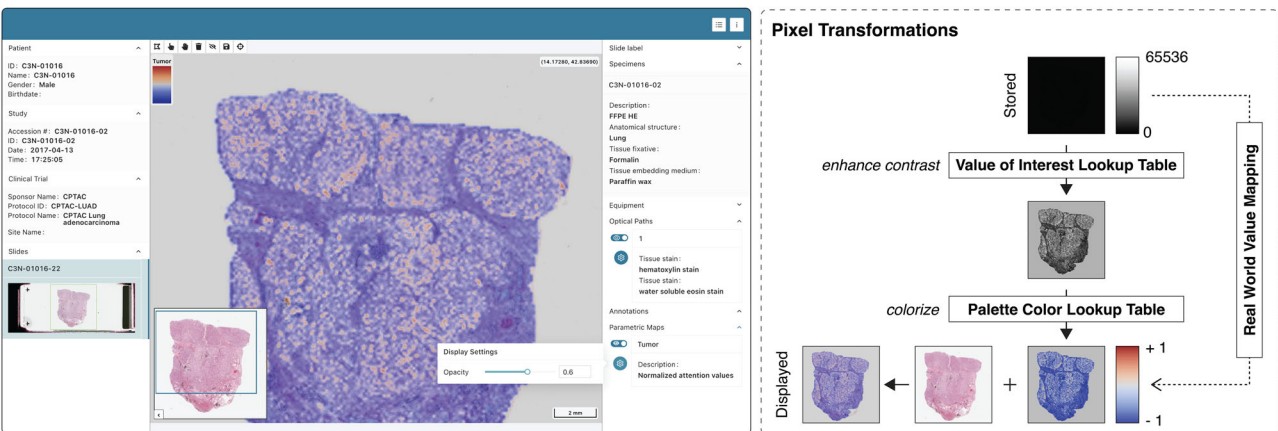

**Fig. 8 | Display of saliency, class activation, or attention maps.** Screenshot of the Slim user interface displaying an attention map image overlaid on the source slide microscopy images from Clinical Proteomic Tumor Analysis Consortium (CPTAC) project (upper left). The attention map was derived from the slide microscopy image using an attention-based image classification model. Note that the pixel data of the parametric map are colorized by the viewer using the palette color lookup table that was embedded into the image object by the model developer.

Each measurement is further associated with a unit, which is coded using the Unified Code for Units of Measure (UCUM) coding scheme. To accommodate different image annotation tasks, Slim can be configured with a set of codes for relevant names and their corresponding values, which are then made available to users via the UI to allow them to annotate image regions using a constraint set of relevant terms (Fig. 6c). Before a user starts drawing ROIs, Slim presents the configured names to the user as questions and prompts them for the corresponding values as answers, either by providing configured categorical value options via a dropdown menu from which the user can select the appropriate value (Fig. 6c) or an input field into which the user can insert a numerical value. To this end, Slim only displays the meaning of coded names, values, and units via the UI to facilitate interpretation by human readers, but internally stores the data as machine readable code triplets, consisting of a code value, a coding scheme designator, and a code meaning (Fig. 6d). When a user ultimately saves a collection of ROI annotations, the graphic data and associated measurements and qualitative evaluations of each ROI are encoded in a DICOM Comprehensive 3D SR document using SR template "Planar ROI Measurements and Qualitative Evaluations" and the complete SR document is stored via DICOMweb. The SR template is extensible and allows for the inclusion of an unlimited number of qualitative evaluations or measurements per ROI using a multitude of coding schemes, thereby providing sufficient flexibility to support a wide range of projects and annotation use cases, while also providing a well-defined schema to facilitate the collection of structured, machine-readable annotations that can be readily used for ML model training or validation[17,22,26]. Notably, the same template is used for capturing annotations and measurements for radiology images[22,26] and can also be used for storing ML model outputs[26].

**Visual interpretation of computational image analysis results**
Slim employs different pixel transformations for display of vector and raster graphics. Spatial coordinates stored in DICOM Comprehensive 3D SR and Microscopy Bulk Simple Annotations instances are colorized using a RGB color that is set by the user and are then superimposed onto the source images as a separate vector layer (Fig. 7). Users have the ability to interactively select individual ROIs to inspect associated measurements and qualitative evaluations and can colorize ROIs based on a selected measurement to assess the distribution of measurement values in spatial tissue context. Pixel data stored in Segmentation and Parametric Map objects are rescaled

and colorized using a VOI LUT and palette color LUT, respectively, analogous to grayscale VL Whole Slide Microscopy Image objects. However, the resulting pseudocolor images are superimposed on the slide microscopy images, and depending on the opacity chosen by the user, overlaid images are blended with underlying images using alpha compositing (Figs. 7, 8). Of note, Segmentation and Parametric Map objects also allow for the inclusion of VOI and palette color LUTs, enabling the developer or manufacturer of the image analysis algorithm to specify exactly how algorithm outputs should be presented to human readers for visual interpretation. This is critical, because the colormap can have a strong effect on the visual assessment of pseudocolor images by human readers[49]. Slim automatically applies provided LUTs to transform the grayscale images and displays a colorbar along with the generated pseudocolor images to aid the reader in interpreting the data. If no palette color LUT is included in an image object, Slim choses a default colormap. In the case of binary or fractional segmentation images, pixels encode class probabilities, which by definition map to real world values in the range from 0 to 1, and Slim thus uses a sequential colormap based on the viridis palette to pseudocolor pixel data of segmentation images by default if no palette color LUT is embedded in the image object. In the case of parametric map images, pixels may encode a variety of quantities, and the relationship between stored values and real world values is not implied but needs to be communicated explicitly. For example, pixel values may be encoded and stored as 16-bit unsigned integers to allow for efficient lossless image compression (e.g., using JPEG-LS codec), but the stored values may represent real world values in the range from −1 to 1, which should be displayed using a diverging colormap. DICOM Parametric Map objects include a real world value LUT to map stored values into real world values. When Slim transforms the pixel data using VOI and palette color LUTs to render pixels as 8-bit color values, it also maps the stored values of interest to the corresponding real world values using the real world value LUT (Fig. 8). Based on the provided real world value mapping information, Slim further determines the lower and upper bounds of the real world data and accordingly choses either a sequential or a diverging colormap. Importantly, Slim does not make any assumptions about the data, but obtains all information pertinent to presentation of the pixel data from the communicated metadata. To facilitate semantic interpretation of the data by human readers, Slim further presents relevant metadata, such as the category and type of segmented or annotated properties as well as the name of the image segmentation or object detection algorithm (Figs. 7, 8).

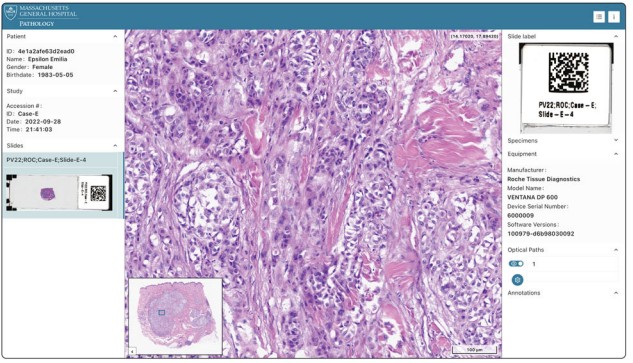
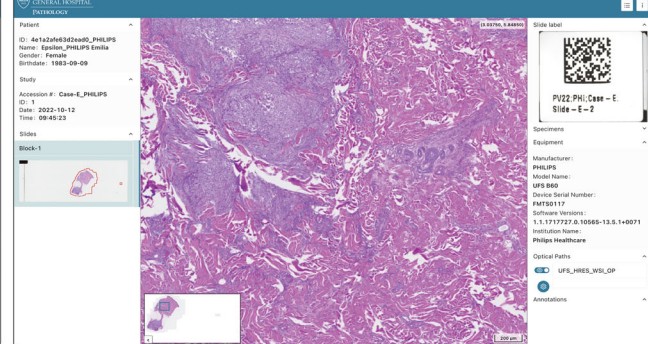

**Fig. 9 | Successful demonstration of interoperability with DICOM conformant scanners and archives from different manufacturers at a DICOM Connectathon.** Screenshots of the Slim user interface displaying slide microscopy images that were acquired and stored by commercially available devices during a DICOM Connectathon at the Path Visions 2022 conference. Shown are images acquired by the Roche Tissue Diagnostics VENTANA DP 600 scanner and stored in the J4Care SMooTH Archive (left) as well as images acquired by the Philips Healthcare UFS B60 scanner and stored in the Sectra Medical VNA (right).

## Connecting with multiple DICOMweb servers to support multi-reader studies and machine learning model inference workflows

There are situations in which it is not desirable to store image annotations or image analysis results alongside the source images. For example, when collecting annotations from users during a controlled, blinded multi-reader study, one may not want to give individual readers access to each other's annotations. Similarly, when evaluating different machine learning models, one may want to persist the outputs of each model independently but nevertheless review the results together with the source images without having to duplicate the source images. As a way to accommodate these and other advanced use cases, Slim can be configured to connect with multiple DICOMweb servers simultaneously, enabling the viewer to retrieve source slide microscopy images and derived image annotation or analysis results from different data stores via different DICOMweb servers. Similarly, a separate DICOMweb server can be configured for the storage of DICOM SR documents that are created by the viewer. It would even be possible for the DICOMweb servers to reside in different cloud environments, as long as the same OIDC provider can be used for authentication and authorization.

## Achieving interoperability with a wide range of standard-conformant servers

While existing web-based slide microscopy viewers may offer similar visualization or annotation capabilities, Slim provides these capabilities via a standard API and independent of a specific server implementation (Fig. 1). To test the conformity of Slim with the DICOM standard and its interoperability with different DICOMweb server implementations, we participated in several DICOM Connectathons, which have been conducted by the digital pathology community in the United States and Europe since 2017 with broad participation from major digital pathology vendors, academic medical centers, and government agencies[41]. During the Connectathon events that lead up to the Path Visions 2020 and 2022 conferences, Slim achieved interoperability with a total of five commercially available, enterprise-grade vendor neutral archives manufactured by GE Healthcare, Pathcore, Sectra, Naegen, and J4Care as well as with two public cloud services offered by Google and Microsoft (Fig. 9). Slim further successfully demonstrated its ability to display slide microscopy images acquired by different whole slide imaging devices, including slide scanners manufactured by Roche Tissue Diagnostics, Leica Biosystems, 3DHistech, and Philips Healthcare (Fig. 9).

## Discussion

Viewing of slide microscopy image in digital pathology is starting to become a clinical reality[55] and AI/ML-based analysis of slide microscopy images in computational pathology is opening new avenues for biomarker development, clinical decision support, and computer-aided diagnosis[3,4,6,56]. Furthermore, with the advent of highly multiplexed tissue imaging and its unique ability to provide insight into phenotypic states and disease manifestations at the single-cell level in spatial tissue context, slide microscopy is also increasingly finding applications in basic research[1,57–59]. Despite the great potential for novel imaging techniques and computational image analysis methods to transform biomedical research and in vitro diagnostics, the lack of standardization of the imaging data and the limited interoperability between different image acquisition, storage, management, display, and analysis systems have been impeding the reproducibility of image-based research findings, the re-use and combination of images and image-derived data across research projects, and the clinical integration of digital and computational pathology[6,11,25,28,41,60–62]. The digital pathology community has long recognized these problems and has invested into the development of the DICOM standard for storage and communication of slide microscopy images[24,63]. In recent years, DICOM has seen increasing adoption by industry and academia[25,41] and is promoted by several professional societies and associations[6,28,60–62]. DICOM has further turned out to be useful for the management of slide microscopy imaging data in biomedical imaging research[25,26,38,64] and has been adopted by several national and multi-national initiatives for biomedical imaging research and development, including the IMI BigPicture[30] and CHAIMELEON[65] in Europe and the NCI IDC[29] in the United States.

The mission and vision of the IDC is to enable multi-modal imaging science in the cloud and enable the visualization and analysis of collections of heterogeneous imaging data from different imaging modalities across domains and disciplines in a common cloud environment[29]. The use of DICOM as a common data standard has proved key for realizing this vision and the DICOM conformity of Slim has made it possible to leverage a common digital imaging infrastructure and platform that is shared with the radiology viewer[29,34]. The reliance on the DICOM standard has further enabled the digital pathology and radiology communities to jointly develop and maintain common libraries and tools for the communication and visualization of biomedical imaging data in DICOM format in a device-independent and modality-agnostic manner[25,29,34]. Furthermore, since DICOM also defines information objects for derived image annotations and image analysis results, the use of DICOM has advantages beyond image display and enables a unified approach to imaging data management across modalities, projects, domains, departments, and institutions to promote interdisciplinary imaging data science and integrated diagnostics[26].

While the digital pathology community has been focusing on the development and adoption of the DICOM standard, the microscopy

research community has been investing into the Open Microscopy Environment (OME)[66–69]. Given its reliance on DICOM, Slim does not directly support OME. However, microscopy images stored in OME-TIFF files[66] can generally be readily converted into DICOM files without loss of information and can then be stored, queried, and retrieved via DICOMweb. For the IDC, we converted all microscopy image files, including OME-TIFF files, into so-called dual-personality DICOM-TIFF files, which have both a DICOM and a TIFF header and can thus be read by DICOM and TIFF readers without duplication of the pixel data[70]. We further contributed to the development of the widely-used Bio-Formats library[71] to add support for reading and writing DICOM files [Bio-Formats release notes[72]] and thereby facilitate the use of DICOM in basic research. It is also important to note that Slim relies on DICOMweb services rather than on DICOM files. This distinction is subtle but important, because DICOMweb is a network API specification and is thus only concerned with how data is communicated via messages using the hypertext transfer protocol (HTTP) and does not impose any constraints on how data is stored by server applications. Therefore, it would in principle be possible to expose data that is stored server side in OME format (either in the form OME-TIFF files[66] or OME-NGFF objects[73]) as DICOMweb resources, by transcoding them on-the-fly as long as the metadata that is required for DICOMweb resources is available and compatible. We recognize the importance of OME in the research setting and are convinced that DICOM and OME can and should coexist. Therefore, we are closely collaborating with the OME community and are actively contributing to the development of OME and the harmonization of image metadata between DICOM and OME[11] to maximize interoperability between different devices and tools.

There are a growing number of commercially available slide scanners from different manufacturers (e.g., Roche Ventana DP 200, Leica Aperio GT 450, 3DHistech PANNORAMIC) that can output brightfield as well as fluorescence microscopy image data directly in DICOM format. Despite the increasing adoption of DICOM in digital pathology, several slide scanners unfortunately continue to output data in proprietary formats, and therefore currently DICOM conversion may be required[25]. To address this issue, several open-source tools are available to convert a variety of proprietary formats into standard DICOM format[38,70,74,75]. Importantly, enterprise medical imaging workflows nowadays evolve around DICOMweb and are increasingly moving from on premises into cloud[28,41,42,76]. Accordingly, major public cloud providers have started to offer cloud-based DICOMweb services (e.g., Cloud Healthcare API on Google Cloud Platform, DICOM service on Microsoft Azure, and HealthLake Imaging on Amazon Web Services), which offer scalable storage, management, and communication of imaging data in DICOM format as a service and have the potential to fundamentally reshape data governance at healthcare enterprises and research institutes around the world. Slim is able to interoperate with existing DICOMweb services in the cloud and supports OIDC-based authentication and authorization.

In the case of the IDC, Slim interacts with the DICOMweb services provided by the Cloud Healthcare API of the Google Cloud Platform[29]. It is worth noting that we had no control over the design or implementation of the Google product and developed Slim independently in a local development environment using different open-source DICOMweb servers. The fact that Slim nevertheless achieves interoperability with the Cloud Healthcare API is due to the fact that both systems conform to the standard DICOMweb API specification and serves as a testament to the power of standardization. Through participation in multiple DICOM Connectathons, Slim has further repeatedly demonstrated its ability to interoperate with many other image storage and communication systems from different manufacturers. By relying on the internationally accepted standard for medical imaging, Slim enables scientists to leverage existing medical-grade infrastructure and a wealth of existing software tools[33] for

quantitative tissue imaging research and allows for visualization of images, image annotations, and image analysis results independent of the device that is used to store and manage the data as well as the device that acquired or generated the data (slide scanner, annotation tool, ML model, etc.). To the best of our knowledge, Slim is the first and currently only open-source web viewer that enables the use of DICOMweb services for interactive visualization and annotation of brightfield and fluorescence slide microscopy images as well as of derived image analysis results. While other web viewers were described in the literature to have the ability to display slide microscopy images stored in DICOM files[38,64,77], these viewers still rely on custom network APIs for querying and retrieving image data from a server over the web and therefore remain tightly coupled to specific server implementations. Furthermore, they depend on the browser's JavaScript API for client-side decoding and rendering of image tiles and therefore require a specialized tile server to transcode the stored pixel data into a simple representation and format (e.g., PNG) that can be directly decoded and rendered by web browsers. In contrast, Slim can be connected with any server that exposes a DICOMweb API and, by handling the decoding, transformation, and rendering of image tiles client side in a browser-independent manner, enables the server to send the pixel data as it is stored.

A number of design decisions in Slim reflect the current state of client-server network connections and computing capacities of the systems available to our target users. First, while networks are efficient at bulk data transfers, round-trip latencies for repeated rendering requests tend to be less efficient than local computation when the client machine has the power to perform the calculations. Second, modern client machines are generally equipped with powerful processors and hosting servers with similar compute resources is expensive. Third, performing compute- and memory-intensive operations on the server may work well for a few users but is difficult and costly to scale to a large number of users. In the face of these considerations, Slim performs all image decoding, transformation, and rendering operations client side in the browser, thereby placing low functional requirements on DICOMweb servers and increasing the likelihood of achieving interoperability with different servers. Client-side processing further has the advantages that the total computational load is distributed across clients, leveraging each client's processing units, and that pixel data need to be retrieved only once per client but can subsequently be repeatedly transformed locally without additional server round trips to enable highly interactive visualization and a smooth viewing experience. To maximize the performance of client-side processing, Slim utilizes state-of-the-art web technologies. Specifically, all image decoding, transformation, and rendering operations are either implemented in C/C++, compiled to WebAssembly, and run in parallel on the central processing unit (CPU) using multiple web workers, or implemented in OpenGL Shading Language (GLSL) and run in parallel on the graphical processing unit (GPU) using WebGL. The use of WebAssembly not only improves performance, but also allows Slim to reuse well established C/C++ libraries for decoding compressed pixel data and color management without having to rely on a particular browser's implementation. This is important, because some slide scanners make use of advanced image compression algorithms and color space transformations (e.g., JPEG 2000 with reversible color transform and multiple component transformation) that can better preserve image quality but are not widely used on the web and are hence not well supported by all browsers. By providing its own browser-independent decoders and transformers, Slim facilitates consistent image quality across different browsers.

Achieving interoperability through standardized metadata and standard APIs is important for image visualization and annotation, but maybe even more so for the development of ML models using heterogenous clinically acquired imaging data from different institutions as well as for performing ML model inference and integrating inference

results into clinical workflows at different institutions[6,18,26]. Each institution or even department may use a distinct set of devices for acquisition, storage, or management of imaging data and realization of ML in the real world thus hinges on the ability of ML models to programmatically find, access, interoperate, and (re)use data[2,78]. Slim can support ML workflows through collection of image annotations from expert readers that can be readily used as ground truth for model training and validation and through display of computational image analysis results to expert readers upon model inference. Since Slim stores image annotations and retrieves image analysis results in DICOM format over DICOMweb, it facilitates the decoupling of ML models and viewers[26] and the integration of outputs from different ML models.

In conclusion, DICOM is a practical standard for quantitative tissue imaging and Slim is an interoperable web-based slide microscopy viewer and annotation tool that implements the standard and opens new avenues for standard-based biomedical imaging data science and computational pathology. The viewer application and its underlying libraries are freely available under a permissive open-source software license: https://github.com/ImagingDataCommons/slim.

# Methods

## User interface design and user flow

Slim's user interface (UI) is designed for slide microscopy workflows in digital and computational pathology and consists of three core components: i. the Worklist component, which lists available imaging studies (cases) and allows users to dynamically filter and select studies for display; ii. the Case Viewer component, which displays metadata about the patient and study, lists the slides that are contained within the study (case), and displays the overview image of each slide; and iii. the Slide Viewer component, which displays volume and label images of the currently selected slide as well as descriptive metadata of the imaged specimen(s) as well as the image acquisition equipment and context. The Case Viewer component searches for slide microscopy images within a selected DICOM Study and groups matched images into digital slides. A digital slide hereby constitutes the set of images that were acquired of a tissue specimen within the same image acquisition context. The images of a given slide may include multiple channels or focal planes and generally correspond to one DICOM Series. Upon selection of a slide, Slim routes the user to the Slide Viewer component, which retrieves the metadata for all image instances of the selected slide and creates the main viewport for the display of the volume images, which contain the tissue on the slide, a separate viewport for the display of the associated label image, and panels for display of relevant specimen, equipment, and optical path metadata that are shared amongst the images. The Slide Viewer component also automatically searches for available image annotations and analysis results that share the same frame of reference as the volume images and makes the data available for display to the user. Alternatively to this user flow, a user or a device can also directly navigate to a particular case or slide using a study- or series-specific unique resource locator (URL) that routes the application to the Case Viewer or Slide Viewer, respectively.

## Data management and communication

All data communication between Slim and origin servers is performed using standard transactions of the DICOMweb Study Service. The transactions are also known as RESTful services (RS). Specifically, the application queries the server for available DICOM studies, series, or instances using the DICOMweb Search Transaction (QIDO-RS), fetches instance metadata and bulkdata (e.g., image frames) using the DICOMweb Retrieve Transaction (WADO-RS), and stores created annotation instances using the DICOMweb Store Transaction (STOW-RS)[25]. Metadata obtained as Search Transaction Resources (DICOM Part 18 Section 10.6.3.3) and Retrieve Transaction Metadata Resources (DICOM Part.18 Section 10.4.1.1.2) are retrieved in JavaScript Object Notation (JSON) format using media type "application/dicom+json". The JSON documents are structured according to the DICOM JSON Model (DICOM Part 18 Chapter F). Individual image frames obtained as Retrieve Transaction Pixel Data Resources (DICOM Part 18 Section 10.4.1.1.6) are retrieved either uncompressed using media type "application/octet-stream" or JPEG, JPEG 2000, or JPEG-LS compressed using media types "image/jpeg", "image/jp2" and "image/jpx", or "image/jls", respectively. Other bulkdata (e.g., ICC profiles or vector graphic data) are retrieved uncompressed using media type "application/octet-stream".

## Software implementation

Slim is implemented as a single-page application in TypeScript using the React framework. The application has a modular architecture and viewer components are implemented as independent, reusable React components. Complex low-level operations are abstracted into separate JavaScript libraries, which are maintained and developed by the authors in collaboration with open-source community members. Specifically, the Slim application relies on the dicomweb-client, dcmjs, and dicom-microscopy-viewer JavaScript libraries[25] to interact with DICOMweb servers over HyperText Transfer Protocol (HTTP), to manage DICOM datasets in computer memory, and to decode, transform, and render images into viewport HyperText Markup Language (HTML) elements, respectively. For rendering raster and vector graphics in the viewport, the dicom-microscopy-viewer library internally uses the OpenLayers JavaScript library, a well-established and well-tested library for building geographic mapping applications.

Slim is aware of the multi-scale pyramid representation of tiled whole slide images, which it determines from the structural metadata of DICOM VL Whole Slide Microscopy Image instances. Based on the image metadata, the viewer precomputes the tile coordinates and then dynamically retrieves the necessary image frames from the server using the Frame Pixel Data resource of the DICOMweb Retrieve Transaction (DICOM Part 18 Section 10) when the user zooms or pans and then subsequently decodes, transforms, and renders each frame in the viewport for display. The viewer supports both the TILED_FULL and TILED_SPARSE dimension organization type and additionally supports concatenations, where an individual large image instance may be split into several smaller instances that are stored and communicated separately. Of note, the viewer expresses the intent to retrieve the frame pixel data in their original transfer syntax by specifying a wide range of acceptable media types via the accept header field of DICOMweb request messages.

Image frames may be served by the DICOMweb server in a variety of image media types and Slim decodes the pixel data client-side using WebAssembly decoders. The operations for decoding of JPEG, JPEG-2000, JPEG-LS compressed image frames are implemented in C++ using the established libjpeg-turbo, OpenJPEG, and CharLS C/C++ libraries, respectively. The C++ implementations are compiled into WebAssembly using the Emscripten C++ library and exposed via a JavaScript API. The decoding operations are performed in the browser in parallel using multiple Web Worker threads/processes.

Slim implements various DICOM pixel transformation sequences (DICOM Part 4 Chapter N). The operations of the DICOM Profile Connection Space Transformation for true color images are implemented in C++ using the Little-CMS C library, compiled into WebAssembly using the Emscripten C++ library, and run on the CPU immediately after the decoding operations in the same Web Worker thread/process for maximal memory efficiency. The operations of the DICOM VOI LUT Transformation and Advanced Blending Transformation for grayscale and pseudocolor images are implemented as graphic shaders in the OpenGL Shading Language (GLSL) and are run on the GPU using WebGL. All pixel data transformations are performed in memory using 32-bit floating-point value representation.

The Slim annotation tools enable users to draw, label, and measure regions of interest (ROI) on displayed images using DICOM Structured Reporting (SR). The points of user-drawn vector graphic geometries are recorded as 3D spatial coordinates (SCOORD3D) in the reference coordinate system, which are independent of pixel matrices of specific image instances and apply to all resolution levels of the multi-resolution pyramid. Slim supports the following graphic types defined by the standard for the SCOORD3D graphic type (DICOM Part 3 Chapter C): POINT, POLYLINE, POLYGON, ELLIPSE, and ELLIPSOID. Users can save their annotations as DICOM Comprehensive 3D SR documents, which are structured according to SR template TID 1500 "Measurement Report"[22]. Individual user-drawn ROI annotations are included in the report document content using SR template TID 1400 "Planar ROI Measurements and Qualitative Evaluations". The SR templates are implemented in JavaScript in the dcmjs JavaScript library.

When the user draws onto the images using the annotation tools, the viewer internally tracks and renders the graphic data as 32-bit floating-point values in the form of two-dimensional coordinates defined relative to the total pixel matrix of the highest resolution image. When the user stores the data in a Comprehensive 3D SR document, the viewer maps the two-dimensional image coordinates at sub-pixel resolution into three-dimensional slide coordinates at millimeter resolution as required by SCOORD3D graphic types. To this end, the viewer constructs an affine transformation matrix from image metadata using equations defined in the standard (DICOM Part 3 Chapter C) and performs the necessary translation, rotation, and scaling of coordinates using a single matrix multiplication. The metadata of the highest resolution image are used to keep potential inaccuracies due to floating-point arithmetic at a minimum.

When Slim routes to the Slide Viewer component, it also searches for existing image annotations and analysis results in the form of i. DICOM Comprehensive 3D SR instances structured according to template TID 1500 "Measurement Report" containing spatial coordinates, qualitative evaluations, and measurements for a small number of ROIs (e.g., tumor regions); ii. DICOM Microscopy Bulk Simple Annotation instances containing spatial coordinates and measurements for annotation groups with a potentially very large number of ROIs (e.g., individual cells or nuclei); iii. DICOM Segmentation instances containing binary or probabilistic segmentation masks; and iv. DICOM Parametric Map instances containing saliency, activation, or attention maps or other heatmaps. The viewer then parses the objects and creates a UI component for each discovered annotation (content items of SR template TID 1400 "Planar ROI Measurements and Qualitative Evaluations" in the SR document content), annotation group (item of the Annotation Group Sequence attribute), segment (item of the Segment Description Sequence), and parameter mapping in the side panel.

Segmentation and Parametric Map images are grayscale images and their pixel data are transformed analogously to the pixel data of grayscale slide microscopy images (see above).

To facilitate overlay of SCOORD3D graphic data onto the image pixel data, the viewer maps the three-dimensional slide coordinates to two-dimensional image coordinates, using the highest resolution image as reference. To this end, the viewer constructs an affine transformation matrix from the metadata of the highest resolution image, which represents the inverse of the transformation described above in "Generation and storage of user-drawn image annotations". The affine transformation matrix is constructed only once and then cached to avoid repeated matrix inversion and thereby speed up the mapping of coordinates into the total pixel matrix.

### Authentication and authorization

Slim utilizes the OpenID Connect (OIDC) protocol for authenticating users and authorizing the application to access secured DICOMweb services based on the OAuth 2.0 protocol. By default, Slim uses the OAuth 2.0 Authorization Code grant type with Proof Key for Code Exchange (PKCE) challenge. In case this grant type is not supported by an authorization server, Slim can be configured to use the legacy Implicit grant type instead. To enable authentication and authorization, Slim needs to be configured with the URL of the identity provider, the OIDC client ID, and a set of OIDC scopes. Slim will then authenticate users with the identity provider and obtain personal information about the user such as their name and email address. This information will automatically be incorporated into stored DICOM SR documents along with image annotations generated by the user to enable downstream applications to identify the person that made the observations. Slim will further request a token from the authorization server to access the configured DICOMweb endpoint and will include the obtained access token into the header of DICOMweb request messages.

### Imaging data sets

To demonstrate the display capabilities of Slim, several public DICOM imaging collections of the IDC were used: i. brightfield microscopy images of hematoxylin and eosin (H&E) stained tissue sections from The Tumor Genome Atlas (TCGA) and the Clinical Proteomic Tumor Analysis Consortium (CPTAC) projects; ii. fluorescence microscopy images of tissue sections stained with DAPI or Hoechst and multiple fluorescently-labeled antibodies via cyclic immunofluorescence imaging (CyCIF) from the Human Tissue Atlas Network (HTAN) project. The DICOM files of these images are publicly available via the IDC portal at https://imaging.datacommons.cancer.gov. Images obtained by the IDC were originally stored as TIFF files, structured based on either the proprietary SVS format (i) or the Open Microscopy Environment (OME) format (ii), and were converted into standard DICOM format for inclusion into the IDC using the PixelMed Java toolkit[70]. In brief, pixel data were copied from the original TIFF files into DICOM files, carefully avoiding lossy image re-compression, and structural image metadata (image height and width, pixel spacing, photometric interpretation, etc.) as well as information about the microscope (manufacturer, model name, software versions, etc.) were copied from TIFF fields into the corresponding DICOM data elements. Additional clinical and biospecimen metadata were obtained from different sources in a variety of formats and were harmonized to populate the appropriate DICOM data elements. Specifically, available information about the specimen preparation (collection, fixation, staining, etc.) were described via the DICOM Specimen Preparation Sequence attribute using DICOM SR template TID 8001 "Specimen Preparation". Further details on the conversion process are provided at https://github.com/ImagingDataCommons/idc-wsi-conversion. Additional collections of DICOM slide microscopy images were used, which were directly stored in DICOM format by commercially available slide scanners. The DICOM files of these images are publicly available via the NEMA File Transfer Protocol (FTP) server at ftp://medical.nema.org.

To demonstrate the annotation capabilities of the viewer, regions of interest (ROIs) were manually demarcated in brightfield and fluorescence microscopy images and anatomic or morphologically abnormal structures were described using SNOMED CT codes. Since Slim is used in the IDC for cancer research, the focus was placed on the classification of tumors and related tumor findings, such as the associated morphology and topography. To this end, a Slim configuration file was created that included the SNOMED CT codes for tumor morphology and topography pertinent to the tumor types of TCGA and CPTAC collections (Supplementary Software). A DICOM Correction Proposal (CP) was submitted by the authors to include these sets of codes into the DICOM standard (CP 2179) and thereby make them available for research and commercial use worldwide without licensing restrictions.

To demonstrate the features of the viewer for visualization and interpretation of image analysis results, existing, publicly available data

sets were used. Specifically, a public collection of pan cancer nuclei segmentation results[79] was obtained from The Cancer Imaging Archive (TCIA) and converted from custom CSV format into standard DICOM format using the highdicom Python library[26]. Contours of detected nuclei and associated nuclear size measurements were encoded in the form of DICOM Microscopy Bulk Simple Annotations objects. Additionally, semantic segmentation masks were derived from the nuclei contours and encoded in the form of DICOM Segmentation objects. To facilitate display of these DICOM objects together with the source slide microscopy images in Slim, references to the source DICOM VL Whole Slide Microscopy Image objects, which were obtained from the IDC, were included into the derived DICOM objects. In addition, all patient, study, and specimen metadata were copied from the source into the derived image objects to enable unique identification and matching. The DICOM Microscopy Bulk Simple Annotations and Segmentation objects are in the process of being included into the IDC and will soon become available via the IDC portal, too.

DICOM Advanced Blending Presentation State objects, which parametrize the display of highly multiplexed immunofluorescence microscopy images from the HTAN project, were generated using the highdicom Python library[26]. For each slide, three presentation states were created for the following three groups of staining targets (compounds investigated) that were used in the majority of imaging studies: 1. DNA, CD45, Pan-Cytokeratin, Vimentin; 2. CD3, CD68, CD20; 3. CD45, E-cadherin, PD1. For each digital slide, the corresponding grayscale images for the given stain were selected based on the coded description of the specimen staining process, which was included in the image metadata (see Imaging data sets above for details), and an optimal value of interest range was determined using the autominerva tool[14]. The DICOM Advanced Blending Presentation State objects have been included in the IDC and are available via the IDC portal.

### Application configuration and deployment

Slim is a single-page application that can be served from a generic static web server and configured to interact with one or more DICOMweb servers. For development and production use of Slim, we have primarily relied on two DICOMweb server implementations: the Google Cloud Healthcare API, which is a public cloud service of the Google Cloud Platform (GCP), and the DCM4CHE Archive[40], which is a free and open-source Java application that can be readily installed and deployed on different operating systems and machines.

For local development, an NGINX web server was used to serve the Slim single-page application as a static website and the DCM4CHE Archive was used as DICOMweb server. Both servers were installed into Linux containers and deployed on localhost using the Docker container runtime. Example Dockerfile and docker-compose.yml configuration files are included in the GitHub repository along with the source code (Supplementary Software).

For development of Slim within the context of the IDC and for production use of Slim via the IDC portal, Google Firebase was used as a static web server and the Google Cloud Healthcare API was used as DICOMweb server. For IDC production use, a reverse proxy server was used, because the Google Cloud Healthcare API requires authorization to access DICOMweb endpoints. The reverse proxy redirects DICOMweb request messages from the viewer to the Google Cloud Healthcare API while imposing limits on the aggregate and IP-specific traffic in order to place a cap on network egress and DICOM operation charges[29]. For development and testing, additional Slim instances were hosted on GCP using the Google Identify and Application Management service as an identity provider and authorization server. To enable access to the protected DICOMweb services directly without the use of a reverse proxy, OAuth 2.0 Client ID credentials were created and Slim was configure to use these credentials to authenticate users and to obtain access tokens for access of the secured DICOMweb endpoints of the Google Cloud Healthcare API.

### Reporting summary

Further information on research design is available in the Nature Portfolio Reporting Summary linked to this article.

## Data availability

Imaging data collections from The Cancer Genome Atlas (TCGA), the Clinical Proteomic Tumor Analysis Consortium (CPTAC), and the Human Tumor Atlas Network (HTAN) projects are publicly available via the data portal of the National Cancer Institute's Imaging Data Commons at https://portal.imaging.datacommons.cancer.gov. Imaging data collections from the DICOM WG-26 Pathology Connectathons are publicly available via the NEMA server at ftp://medical.nema.org. To facilitate reproduction of figures, the imaging data that are shown in the paper (subset of the aforementioned imaging data collections) are made available via Figshare[80].

## Code availability

Source code of the software is publicly available on GitHub at https://github.com/ImagingDataCommons/slim and is supplemented by[81]. A copy of the source code is also provided as Supplementary Software.

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

## Acknowledgements

The authors thank Adam von Paternos and Aidan Stein for assistance with deployment of viewer instances on premises and in the cloud. In addition, the authors express their gratitude to Bill Clifford, David Pot, Daniela Schacherer, André Hohmeyer, Ulrike Wagner, Keyvan Farahani and the entire IDC team for thoughtful feedback and discussions and for performing user acceptance testing. The authors are also thankful for the stimulating discussions with Adam Taylor, Robert Krüger, Jeremy Muhlich, Sandro Santagata, Peter Sorger, and other members of the HTAN team regarding the interactive visualization of multiplexed tissue imaging data sets. Special thanks are extended by the authors to members of DICOM Working Group 26 "Pathology" and their support in organizing and conducting the Connectathon events. The results shown here are in part based upon data generated by the TCGA Research Network (http://cancergenome.nih.gov), the Human Tumor Atlas Network (https://humantumoratlas.org), and the National Cancer Institute's Clinical Proteomic Tumor Analysis Consortium (https://proteomics.cancer.gov/programs/cptac). The authors acknowledge the National Cancer Institute and the Foundation for the National Institutes of Health, and their critical role in the creation of the data. This project has been funded in whole or in part with Federal funds from the National Cancer Institute, National Institutes of Health, under Task Order No. HHSN26110071 under Contract No. HHSN2612015000031. The content of this publication does not necessarily reflect the views or policies of the Department of Health and Human Services, nor does mention of trade names, commercial products, or organizations imply endorsement by the U.S. Government.

## Author contributions

C.G. and M.D.H. wrote the initial draft of the manuscript and created the figures. C.G., D.P., I.O., and M.D.H. implemented the software. All authors contributed to the design and testing of the software and the writing of the manuscript.

## Competing interests

All Authors declare the following competing interests: All Authors received funding from the National Institutes of Health for the research. David A. Clunie receives financial compensation as a consultant of Healthcare Tech Solutions (HCTS), as a consultant for Impact Business Information Solutions (IBIS), as a consultant for Mayo Foundation for Medical Education & Research, as a consultant for Essex Leidos CBIIT under under National Cancer Institute Contract No. 75N91019D00024, Task Order 75N91019F00129, as a consultant for Brigham and Women's Hospital NCI Imaging Data Commons (IDC), as a consultant for the University of Leeds Northern Pathology Imaging Co-operative (NPIC), and as a contractor for NEMA as DICOM Editor.
