## [Peer Review File · Nature Communications]

Reviewers' Comments:

Reviewer #1:

Remarks to the Author:

This manuscript entitled "Slim: Interoperable Web Viewer and Annotation Tool for Quantitative Microscopy Tissue Imaging" highlights a new tool for retrieving, visualizing, and interacting with 'slide' microscopy data using DIACOMweb services. The development of multimodal data viewers and analysis tools are highly relevant in the areas of digital pathology and image-based basic research. In particular, the rapid development of highly multiplexed molecular imaging technologies and large-scale tissue imaging/atlasing efforts that are underway require tools like SLIM that are being developed as part of the NCI Imaging Data Commons. The authors' previous iteration of the work lacked technical details and concrete examples of both performance and use cases. However, this revision has been dramatically expanded including well-described examples with brightfield and multiplexed IF data, including demonstrations of image annotation and segmentation. The authors also show direct use of the tool as part of HTAN, a large-scale NIH consortium. Their new discussion on DIACOM vs. OME is welcome and dramatically improves its relevancy to the broader OME community. The authors have sufficiently addressed the reviewers' comments.